



# Technical Note: Modeling Spatial Fields of Extreme Precipitation – A Hierarchical Bayesian Approach

Bianca Rahill-Marier[1], Naresh Devineni[2*] and Upmanu Lall[3]

[1]NCX, New York, NY.

[2]Department of Civil Engineering, City University of New York (City College), New York, NY 10031.

[3]Department of Earth and Environmental Engineering, Columbia Water Center, Columbia University, New York, NY 10027.

*Correspondence to*: Naresh Devineni (ndevineni@ccny.cuny.edu)

**Abstract.** We introduce a hierarchical Bayesian model for modeling spatial rainfall for extreme events of a specified duration that could be used with regional hydrologic models to perform a regional hydrologic risk analysis. An extreme event is defined if any gaging site in the watershed experiences an annual maximum rainfall event, and the spatial field of rainfall at all sites corresponding to that occurrence is modeled. Applications to data from New York City demonstrate the effectiveness of the model for providing spatial scenarios that could be used for simulating loadings into the urban drainage system. Insights as to the homogeneity in spatial rainfall and its implications for modeling are provided by considering partial pooling in the Hierarchical Bayesian framework.

## 1 Introduction

For an existing urban drainage network, a proper consideration of the spatial structure of extreme rainfall events is important for an assessment of the effectiveness of the network for handling urban flooding subsequent to rainfall events of varying duration, especially as concerns emerge as to the resilience of the system under a changing climate. Often, investigators focus on a return period analysis of extreme rainfall at a site considering annual maxima or peaks over threshold for a specific rainfall duration. In a regional context, spatial models of annual maximum rainfall are sometimes considered (Renard et al., 2006; Renard and Lang, 2007; Dyrrdal et al., 2015). However, since the annual maximum is unlikely to always occur for the same event at all sites, these models do not represent the actual structure of potential extreme rainfall events. We address this situation in this note by considering that the rainfall events of interest for a specified duration are ones where any one of the sites in the region experiences an annual maximum event, and the spatial field or rainfall of interest is then the field associated with each such event.

In the exploratory analyses performed for New York City we noted that the structure of storms that lead to annual maximum events at different gages in the region may not be the same and that the basic statistics of rainfall vary across sites. We assemble a dataset of the annual maximum rainfall for each specified duration at each station. Let's denote this as $A_{djt}$, for duration $d$, site $j$ and year $t$. These events may not occur on the same day of each year across the stations. Second, we





consider the rainfall field at all stations associated with the annual maximum at any one station and call this the "spatial field" (SF), $R_{djki}$, where $i$ is identified as an event such that $R_{dkki} = A_{dkt}$ for site $k = j$ for year $t$. $R_{djki}$ then has the rainfall at all sites $j$, for the event where site $k$ has an annual maximum. As a result, the total number of events, $i$ may be much larger than the total number of years of data, $N$. A spatial event field of rainfall is thus conditional on the occurrence of an annual maximum rainfall for any station. This is similar to the temporally concurrent extreme maxima rainfall fields used in Asquith

and Famiglietti (2000). The hierarchical Bayesian models developed consider the spatial field SF with the goal of providing an approach for stochastically generating representative spatial fields of rainfall for a specified duration, such that at least one site in the region experiences an annual maximum event.

In Section 2, we present the data and the context for the application to the Greater New York area. In Section 3, we describe the details of the multivariate hierarchical Bayesian models. The results are discussed in Section 4. Finally, in Section 5, we

present a summary and conclusions.

## 2 Data Description

### 2.1 Greater New York Area Context

The Greater New York City area has high density of man-made infrastructure and hence a complex hydrological landscape. Like many older cities, sanitary and industrial wastewater, and rainwater and street runoff are collected in the same sewers

and conveyed together to treatment plants. Approximately 60 percent of NYC's drainage area is served by these combined sewers. Flooding and combined sewer overflows (CSOs) are a concern, and innovative solutions for using the sewer system itself as flood storage by pumping water to different areas during a storm has been suggested. Such hydrologic system upgrades need to be informed by the spatial variability of extreme precipitation.

There are very few rain gauge records that are longer than twenty or thirty years. Persistent data quality issues further reduce

the available data. This challenge of reconciling sparse data with spatially variable hydrological networks and meteorological phenomena is common to many urban areas. Widely accepted design standards are derived from a set of intensity-duration-frequency curve developed using a daily rainfall record from 1903 to 1951 (NYCDEP 2008). An analysis of precipitation extremes for the region is offered in Wilks and Cember (1993), using daily rainfall data, and McKay and Wilks (1995), using hourly rainfall data. None of these analyses consider the spatial correlation of rainfall.

### 2.2 Precipitation Data

The precipitation data was obtained from the National Climatic Data Center (NCDC 2013). Rain gauges were selected based on the proximity to New York City, data quality, and length of the historical record. Twenty-nine stations were initially identified as lying within a 100-mile radius of Central Park with over 25 years of continuous hourly rainfall records. Sixteen stations were excluded since the resulting data quality was too poor. The final dataset consists of the remaining nine stations

(Table 1); abbreviations for each station used in the figures throughout are provided in the first column.



**[Table 1 - in here]**

**2.3 Diagnostics and Spatial Dependence**

The Shapiro-Wilks test (Shapiro and Wilk, 1965; Royston, 1992) was applied to the log-transformed annual maxima series
at each station for each storm duration from 1-hour to 24-hours. For these 216 time series, the null hypothesis of the
appropriateness of the Log Normal distribution was not rejected for 98% of the sites at the 1% level, 88% at the 5% level,
and 81% at the 10% level. Though other distributions, such as the GEV, Pearson, Log Pearson Type III (LP3) are popular for
extreme precipitation modeling, in our application to the spatial field of rainfall, only one of the sites experiences the annual
maximum event and others may not be extreme values. In such a setting, the Log Normal distribution can often be an
appropriate representation (Raiford et al. 2007), as seen here. Consequently, to illustrate the idea, we consider a lognormal
distribution with spatial correlation across the sites for the NYC example. Other choices could very well be made.

A heat map showing the fraction of annual maximums that occur simultaneously is provided in Figure 1. For these plots, we
define simultaneous storms to be those beginning within +/- n hours of each other (where n is a multiple of the event
duration) to allow for the movement of a storm event over the area, and to identify distinct, independent rainfall events. We
see that precipitation extremes, even within a relatively local area, are frequently not simultaneous. As expected, the
simultaneous fraction of concurrent area increases as the storm duration increases. However, even for a 24-hour duration
less than 60 percent of the events are concurrent. This is true even for the four closest stations – JFK, LGA, Central Park and
Staten Island that are typically used to inform hydrologic design in New York City.

**[Figure 1 - in here]**

This diagnostic analysis highlights the importance of considering the spatial structure of extreme rainfall for an event with a
specified duration.

**3 Methodology**

A hierarchical Bayesian approach that provides the ability to partially pool model parameters across the rain gauge sites was
developed. Full pooling would imply that a parameter (e.g., the mean, variance etc. of the distribution) was homogeneous
across the sites. No pooling would imply that each site is independent. Partial pooling is an intermediate step that allows
information to be shared across sites at a level informed by the data. This results in a multi-level model, where model
parameters are estimated at each site, but are assumed to be drawn from the parameters of distributions that are specified at
the regional level for each parameter (Gelman and Hill 2007). Such an approach has been implemented for



hydrometeorological extremes in Lima and Lall (2010) and Kwon et al. (2008), and for paleoclimate reconstructions by
Devineni et al (2013).

### 3.1 Spatial Fields Hierarchical Model conditioned on the site experiencing an annual maximum

In this model, we consider a conditional process, where site k has experienced an annual maximum event, and the
corresponding rainfall amounts, R$_{djki}$ at all sites are observed. The logarithm of rainfall is considered to be Normally
distributed, and a multivariate Normal distribution is specified for each site k, where an annual maximum has occurred. For
each such condition, we consider partial pooling of the mean rainfall across all sites, and consider the spatial covariance
across sites. We consider that the spatial field of rainfall may actually be different depending on which site experiences an
annual maximum. The hierarchical model is described as below.

$$Y_k \sim MVN(\mu_k,\ \Sigma_k)$$
$$\mu_{kj} \sim N(\omega_k,\ \sigma_k^2)$$

$$Priors$$

$$\Sigma_k \sim Inv - Wishart\ (\Lambda, v)$$
$$\omega_k \sim N(0,1000) \tag{2}$$
$$\sigma_k^2 \sim U(0,100)$$

$Y_k$ is the log of the rainfall field R$_{djki}$ across all sites corresponding to when station *k* has an annual maximum. For the New
York City application, it is a matrix of 64 (years) by 9 (stations) for a given duration, and station k. $Y_k$ is assumed to follow a
multivariate normal distribution with a vector of station means $\mu_k$ and covariance across stations specified by a 9-by-9
matrix $\Sigma_k$. At the second level of the model, the station-specific means $\mu_j$ are assumed to be Normally distributed with a
common mean $\omega_k$ and variance $\sigma_k^2$. This is a partial pooling approach with no covariates, as outlined in Gelman and Hill
(2007). A non-informative conjugate prior, the inverse-Wishart distribution, is assumed for $\Sigma_k$ where $\Lambda$ is the scale matrix
and $v$ is the degrees of freedom (Gelman et al. 2004). If $\Sigma_k$ is a j-by-j matrix, we assume $v$ equivalent to (j + 1) and $\Lambda$ equal
to the j-by-j identity matrix (**I**). This is equivalent to a uniform prior on each variance element of the correlation matrix
(Gelman and Hill 2007). We give $\sigma_k^2$ a non-informative uniform prior, and $\omega_k$ a non-informative conjugate normal prior,
for computational convenience (Gelman and Hill 2007; Gelman et al. 2004).

There are nine stations, and therefore there are nine distinct datasets $Y_k$ and nine distinct models for each storm duration. For
extreme rainfall events, i.e., those that exceed a nominal design return period, we outline a simulation strategy from these
models that pools simulated fields together that represent regional extreme events.





## 3.2 Spatial Fields Single-Level Model

We consider a subset of the previous model where the assumption that the mean log-rainfall is drawn from a common spatial mean is relaxed. This leads to the simpler, no-pooling model represented below.


$$Y_k \sim MVN(\mu_{ks}, \Sigma_{ks})$$
$$\mu_{kjs} \sim N(0, 1000) \qquad (3)$$

$$\Sigma_{ks} \sim Inv - Wishart\ (\Lambda, v)$$

As in the hierarchical model, $Y_k$ is the log of the SF when station $k$ is at an annual maximum. The vector of precipitation means across $j$ stations (including station $k$) is $\mu_{ks}$, with a subscript $s$ to indicate single-level model.

## 3.3 Spatial Fields Simulation for a regional T-year return period

The Spatial Fields model can be used to simulate rainfall fields corresponding to an annual maximum occurring at any one of the sites, k.  Next, if we are interested in design rainfall fields represented by the T-year return period across the domain we follow a two-step process. First, based on the model that is fit, we identify the T year return period annual maximum rainfall even for each site, k. Then from simulations of the multivariate rainfall fields using the model we identify all cases where the

rainfall at site k exceeds the T-year event for that site, and take the corresponding simulated rainfall field across all sites, j. The process is outlined below:

i.   *Threshold Calculation*: For each return period (T) and rainfall duration (D) a precipitation threshold is computed for each station using the posterior mean and variance from the station k's hierarchical Bayesian model. The threshold was computed using frequency factors *K* for the normal distribution (Guo 2006) and the equation below.


$$\log(Y_{T,dk}) = \widetilde{\mu_k} + K(T) * \widetilde{\Sigma_{kk}} \qquad (4)$$

For example, letting k = 1 for Central Park, we compute the SF model from $\mathbf{Y_1}$. We extract $\mu_{11,}$ the Central Park mean and $\Sigma_{11}$, the Central Park variance and use them in equation 4 above.

*ii.*   *Simulate Multivariate field for $Y_k$*: From the hierarchical Bayesian model defined in (2) simulate a large number of realizations M (e.g., equal to 10,000), of the rainfall fields $\mathbf{Y}_k$ corresponding to the case when site k has an annual maximum. These are based on draws from the posterior distributions of the parameters, and hence incorporate a consideration of parameter uncertainty.

iii.  *Extract Subset of Simulations that exceed the T-Year event at site k*:  Retain a subfield $\mathbf{Z}_k$ from $\mathbf{Y}_k$, such that $Y_{dkkm} >$

$Y_{T,dk}$, and m = 1…M,  is the index of the simulation.





iv.  *Pool the T-year return period fields*: The $\mathbf{Z}_k$ are subsamples of rainfall fields from each of the nine models, such that an equal number of draws from each of the k fields is selected. For T=100 years, on average 100 such samples will be generated for M=10000, from each station, and 900 total fields are then available for our application to the New York City data for design or reliability analyses. Note that since there may be multiple sites with annual maxima per event $i$ in the original $R_{djki}$ data, and that these are contained in each random field indexed by k, and we modeled this spatial field, the concurrence of high rainfall at those sites will also be reproduced in the simulations. Similarly, the incidence of high rainfall at multiple stations will also be correctly reproduced across the pooled data across the K simulations.

## 3.4 Model Fitting and Convergence

Fifty-four models (one for each duration and each site) were fit using WinBUGS, using a software extension to R (Lunn et al. 2000; Spiegelhalter et al. 1996). It uses a Markov chain Monte Carlo (MCMC) simulation algorithm (a Gibbs Sampler for the current example), to simulate the posterior probability distribution of parameters. A random normal distribution was used for vector of station means ($\boldsymbol{\mu}, \boldsymbol{\mu_k}, \boldsymbol{\mu_{ks}}$) and a random Wishart distribution was used for the precision. In WinBUGS, the normal distribution is parameterized in terms of precision instead of covariance ($\boldsymbol{\Sigma}, \boldsymbol{\Sigma_k}, \boldsymbol{\Sigma_{ks}}$) as is noted by convention in the model formulas above. We simulated four chains, ran the model for 20,000 iterations and the first half of the simulations were discarded as burn-in.

## 4 Results and Analysis

### 4.1 Bayesian Model Checking

For each model, the convergence of the posterior distribution of each parameter was checked using the shrink factor proposed by Gelman and Rubin (1992) - values under 1.1 for all parameters suggest that the model has converged. For each run 20,000 MCMC iterations, using four chains, were specified. Convergence plots (showing the mixing of the four chains) were visually checked for all cases. All models converged appropriately with each parameter attaining a shrink factor between 1.0 and 1.1, and the large majority reaching 1.0.

We compare the performance of the two SF models using the Deviance Information Criterion (DIC) and pD, recommended *in* Gelman et al. (2004). The scores were virtually identical for the two types of SF models for each rainfall duration. Next, we considered whether the common mean in the hierarchical model converged as successfully as other parameters; it did. Gelman and Hill (2007) suggest that when there are only a small number of groups and the group-level standard deviation is large, multi-level modeling may not add much information. The resulting model will not necessarily perform worse and will



likely resemble the model without pooling (as it does here). The posterior parameters for the resulting simulations are
essentially identical (Figure 2a) and the posterior for the mean only very slightly shrunk (Figure 2b).

**[Figure 2 - in here]**

Next, we consider how the return period thresholds identified in the models compare to observed events. We do so by
plotting the posterior density of the return period event estimated from a sample of 500 simulations and comparing it with
the empirical return period event. Though the Bayesian models can easily simulate any return period (the highest presented
here is 500 years), the reliability of the empirical estimate is dependent on the length of record so it is only reliable as a
goodness of fit measure for shorter return periods. The empirical return period for the ten-year event is estimated as the
accumulated precipitation measurement nearest to the 90th percentile value. Three stations – Central Park, Essex Fells, and
Staten Island – exemplify a range of results (Figure 3).

**[Figure 3 - in here]**

The empirical quantiles for Central Park fall towards the mean of the simulations, the Essex Fells empirical estimate lies
significantly to the left; and the Staten Island empirical estimate lies to the left even of the 5% to 95% range (in blue). Plots
of the remaining stations for 12-hour storm and ten-year return period are provided in Figure A1 of the appendix. The plots
for all nine stations at the 1-hr and 24-hr durations and ten-year return period are provided in Figure A2 and Figure A3
respectively, of the appendix. Differences across stations and across storm durations are not always consistent though some
patterns emerge. For four stations – Central Park, New Brunswick, LGA, and New Milford – empirical quantiles fall within
the 5% to 95% range across storm durations and are even sometimes close to the mean of the simulations (see Figures A1,
A2, and A3 in the appendix for individual plots).

The empirical quantiles for Staten Island are consistently underestimated.  For the 10-yr return period, the empirical estimate
falls below the 5% mark of the simulations for the 1-hr, 12-hr, and 24-hr storms. It's difficult to identify exactly why this
might be the case without significant additional exploratory analysis of the Staten Island data.  However, reducing the return
period to five-years does improve the results with the empirical estimate falling within the 5% to 95% range for the 12-hr
and 24-hr storms, but still outside for the 1-hour storm (Figure 4).

**[Figure 4 - in here]**

This suggests that the distortion in the return period estimate due to the shorter duration of the record is at least partially
responsible for the relatively poor fit of the model. It is important to remember that the simulations reflect additional
information provided by correlating across stations in the model and that an empirical estimate from a short time period,
while one of the only data comparisons we have available to us, has an element of bias and uncertainty as well.



## 5 Summary and Conclusions

For larger cities, a consideration of the drainage network, and the spatial dependence in rainfall at different durations is
important to consider, at least from the perspective of assessing the performance and resilience of the network, and perhaps
also for design considerations. We were interested in formulating and testing a simple model that could directly explore
whether or not, and to what extent there was opportunity to pool regional information on extreme rainfall events to describe
plausible spatial fields of extreme rainfall. This led to postulating and testing a Bayesian model that considers the spatial
field of rainfall associated with an annual maximum occurrence at any site. We considered the application of model to
relatively long rainfall time series from the New York City region. Initial exploratory analyses suggested that the rainfall
characteristics and storm tracks varied by event and by season across the region, such that distinct clusters could be
identified, suggesting that the region had a heterogeneous spatial structure with respect to extreme rainfall (Hamidi et al.
2017). Our applications further clarified the nature of this heterogeneity. It is interesting to also note from the New York City
analysis that there is support for pooling the spatial covariance of rainfall across all sites (irrespective of which one
experienced an annual maximum rain event for a given duration), even though often the exceedance probability distributions
of rainfall for a given duration may differ across sites, even after partial pooling. The hierarchical Bayesian framework
permits a consideration of the uncertainty in parameter and model structure and helps us identify the level of homogeneity
that may be appropriate for representing the processes underlying a particular data set.

Rain gauges are the preferred data source for extreme event modeling because of their long-record, but incorporating radar in
addition to rain gauges could provide the spatial density needed to explore how event rainfall characteristics relate to specific
meteorological phenomena or to provide comparable simulations to existing stochastic models. The radar information would
contain considerably more spatial detail necessary for building the type of model exemplified here. However, radar rainfall
records are much shorter, and consequently, one needs to develop a methodology to appropriately blend the shorter but
spatially richer radar data with the longer but spatially sparse gage data. Our algorithm can be readily applied to a mix of
radar and rain gauge data. However, some extensions need to be pursued to address the very different record lengths of each
data source.

We used a Log Normal distribution applied to rainfall for each duration, to illustrate our approach. The goodness of fit tests
supports this assumption, and this permits some confidence in the kind of conclusions we drew from the applications to the
New York City data. However, other models such as the GEV or Generalized Pareto or other choices for the distribution
could very well be considered. The point here was to highlight the need to consider spatial covariance and an appropriate
blending of local and regional data sources through partial pooling.



## Appendix

### Auxiliary Figures

This appendix includes Figure A1, Figure A2, and Figure A3 which are the plots of the probability distributions of the 1-hr, 12-hr and 24-hr ten-year return period events from the SF models for the nine stations in and around New York City.

### Code Availability

The code for conducting the analysis presented in this paper can be made available upon request.

### Data Availability

Rainfall data for the analysis of the nine sites in and around New York City can be obtained from the public source provided in the references, National Climate Data Center, https://www.ncdc.noaa.gov/data-access/quick-links. The authors can be contacted for any details on the methodology.

### Author Contributions

ND and UL designed the study and edited the manuscript. BRM conducted the analysis and wrote the paper.

### Competing Interests

The authors declare that they have no competing interests.

### Financial Support

This research is supported by the Department of Energy Early CAREER Award No. DE-SC0018124 for the second author and National Science Foundation, Water Sustainability and Climate (WSC) program – award number: 1360446. Partial
support was also provided by the American International Group (AIG) under the "Climate Informed Global Flood Risk Assessment" project.

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

**Table 1: Rain gauge stations in New York City and surroundings**

| Abb. | Location | Latitude | Longitude | Elevation (ft) | Start | End |
|---|---|---|---|---|---|---|
| CP | New York Central Park Observation Belvedere Tower, NY | 40.66889 | -73.9602 | 39.6 | 5/1/1948 | 7/29/2012 |
| EF | Essex Fells Service Building, NJ | 40.8314 | -74.2858 | 106.7 | 7/4/1949 | 8/1/2012 |
| JFK | New York JF Kennedy International Airport, NY | 40.63861 | -73.7622 | 3.4 | 1/1/1949 | 7/29/2012 |
| LGA | New York LaGuardia Airport | 40.77944 | -73.8803 | 3.4 | 5/1/1948 | 7/29/2012 |
| NW | Newark International Airport, NJ | 40.6825 | -74.1694 | 2.1 | 5/1/1948 | 7/29/2012 |
| NB | New Brunswick 3 SE, NJ | 40.4719 | -74.4365 | 26.2 | 6/1/1968 | 2/1/2006 |
| NM | New Milford, NJ | 40.961 | -74.015 | 3.7 | 5/31/1946 | 6/30/1980 |
| SI | New York Westerleigh, NY (Staten Island) | 40.63333 | -74.1167 | 24.4 | 5/1/1948 | 9/1/1992 |
| WT | Watchung, NJ | 40.66222 | -74.4164 | 79.2 | 6/1/1948 | 8/1/2012 |



**Figure 1: Percent of simultaneous or near-simultaneous annual maxima events shown for the site-by-site comparison for nine sites and 1-hr, 6-hr, 12-hr, and 24-hr storms.**







**Figure 2: Density plot of posterior distribution of (a) simulations and (b) mean parameters for Central Park 12-hour hierarchical and non-hierarchical SF model. Posterior means and simulations are shown on an untransformed scale (i.e., the mean is log mean).**




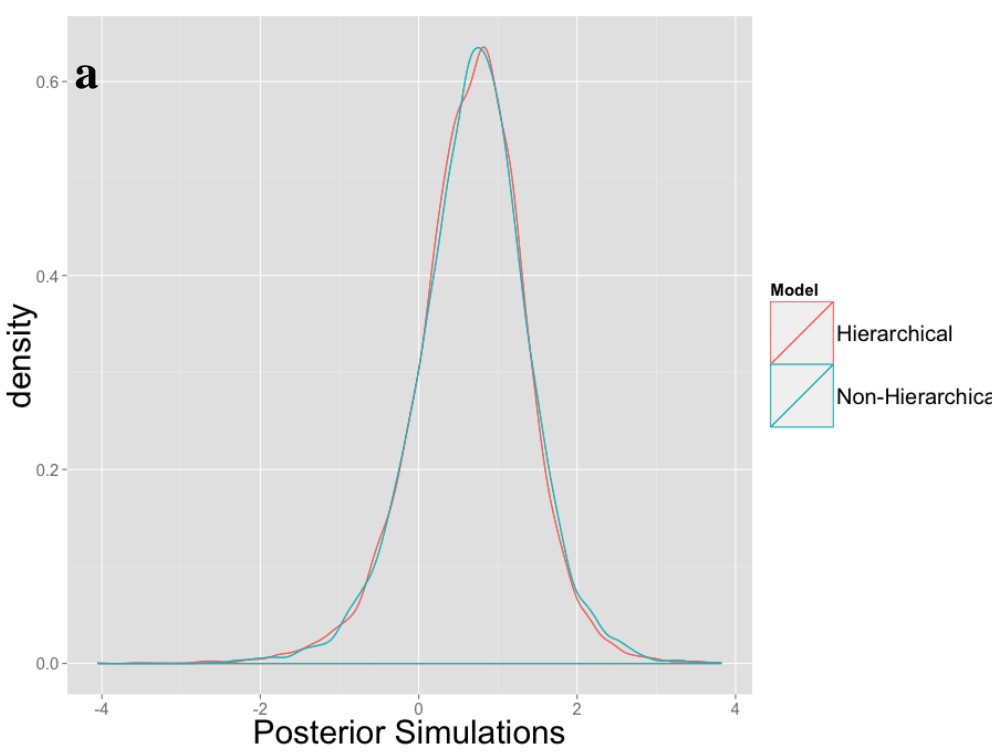




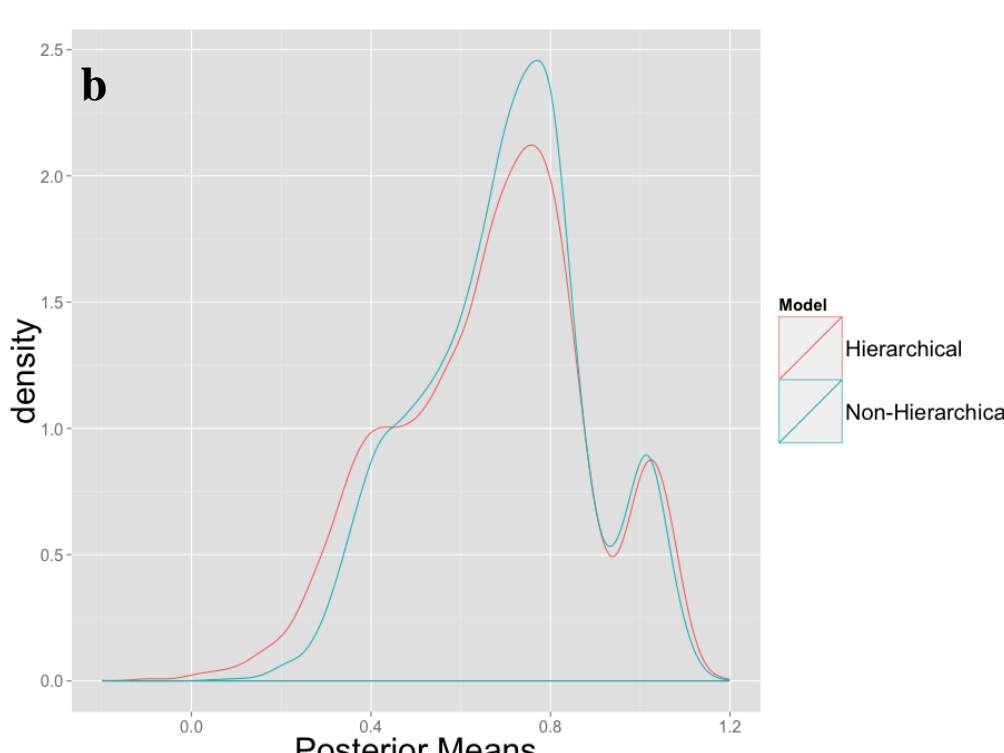






**Figure 3: Density plots of the 12-hr ten-year return period event from SF models for Central Park (top), Essex Fells (center), and Staten Island (bottom). The empirical 10-year**

**event is the dotted line and the 5% - 95% range of simulations is shaded in blue.**


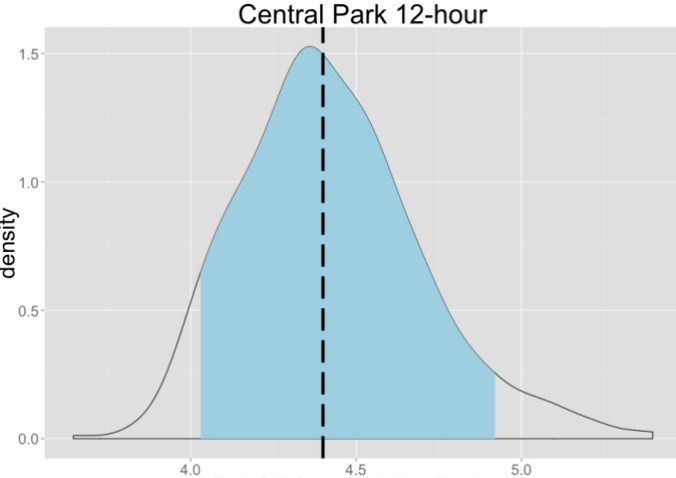



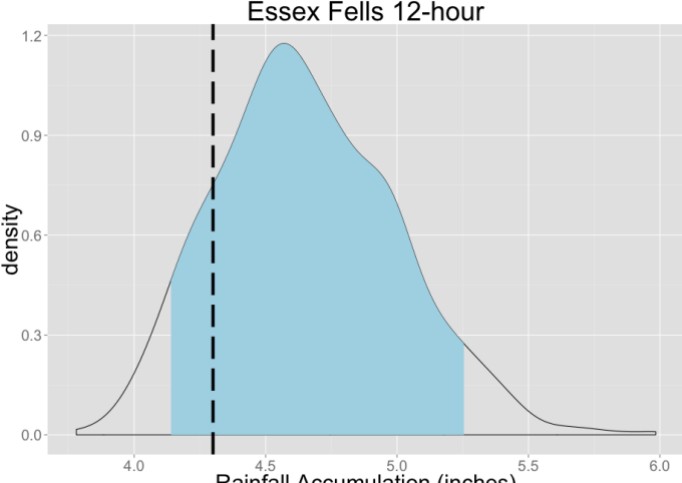


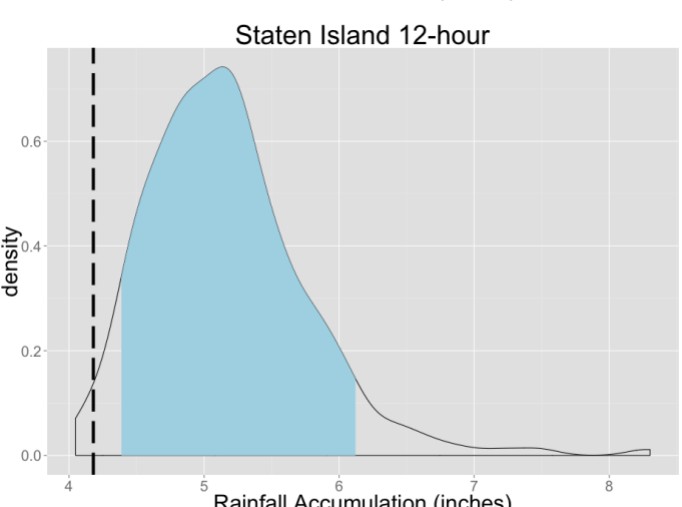



**Figure 4: Density plots of 5-year return period events for 24-hour (top), 12-hour (middle) and 1-hour (bottom) storms from Staten Island SF model simulations. The empirical 5-year event is shown as a dotted line and the 5% - 95% range of the simulations is shaded in blue.**

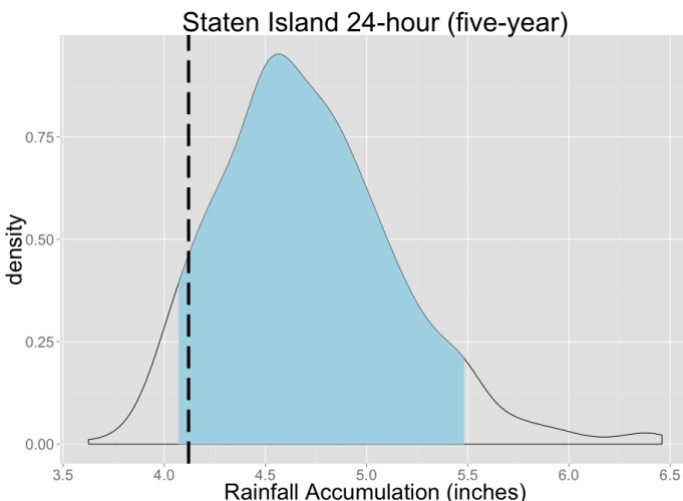

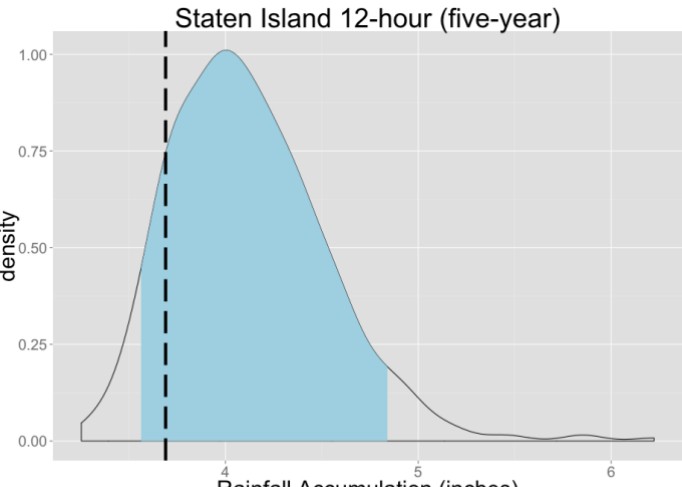

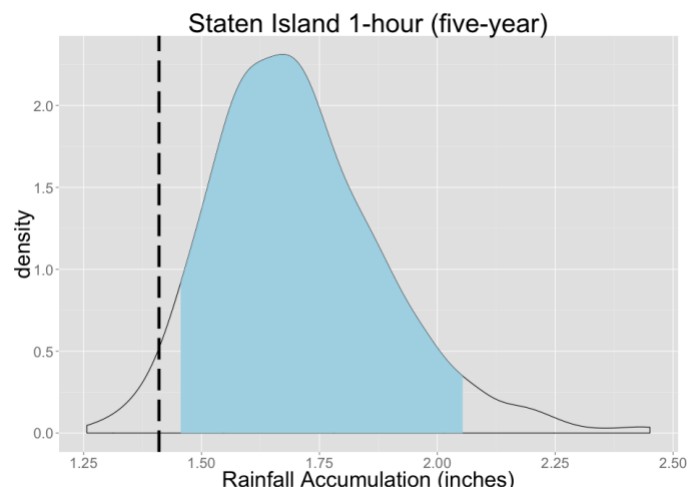

415

**Figure A1. Density plots of the 12-hr ten-year return period event from SF models for the other six stations (JFK, LGA, New**

445 **Brunswick - top to bottom, left panel; Newark, New Milford, Watchung - top to bottom, right panel). The empirical 10-year event**

**is the dotted line, and the 5% - 95% range of simulations is shaded in blue.**



450

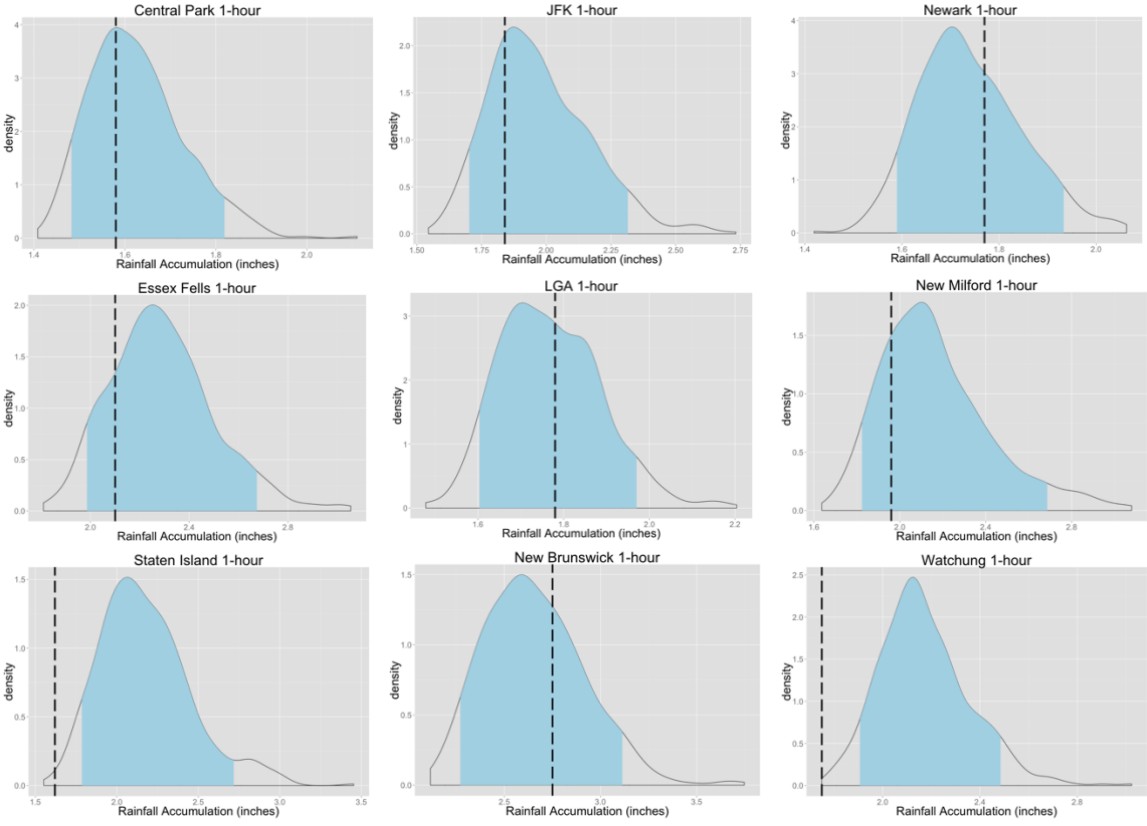

**Figure A2. Density plots of the 1-hr ten-year return period event from SF models for the nine stations (Central Park, Essex Fells, Staten Island – top to bottom left panel; JFK, LGA, New Brunswick - top to bottom, middle panel; Newark, New Milford, Watchung - top to bottom, right panel). The empirical 10-year event is the dotted line, and the 5% - 95% range of simulations is shaded in blue.**

455

460

465



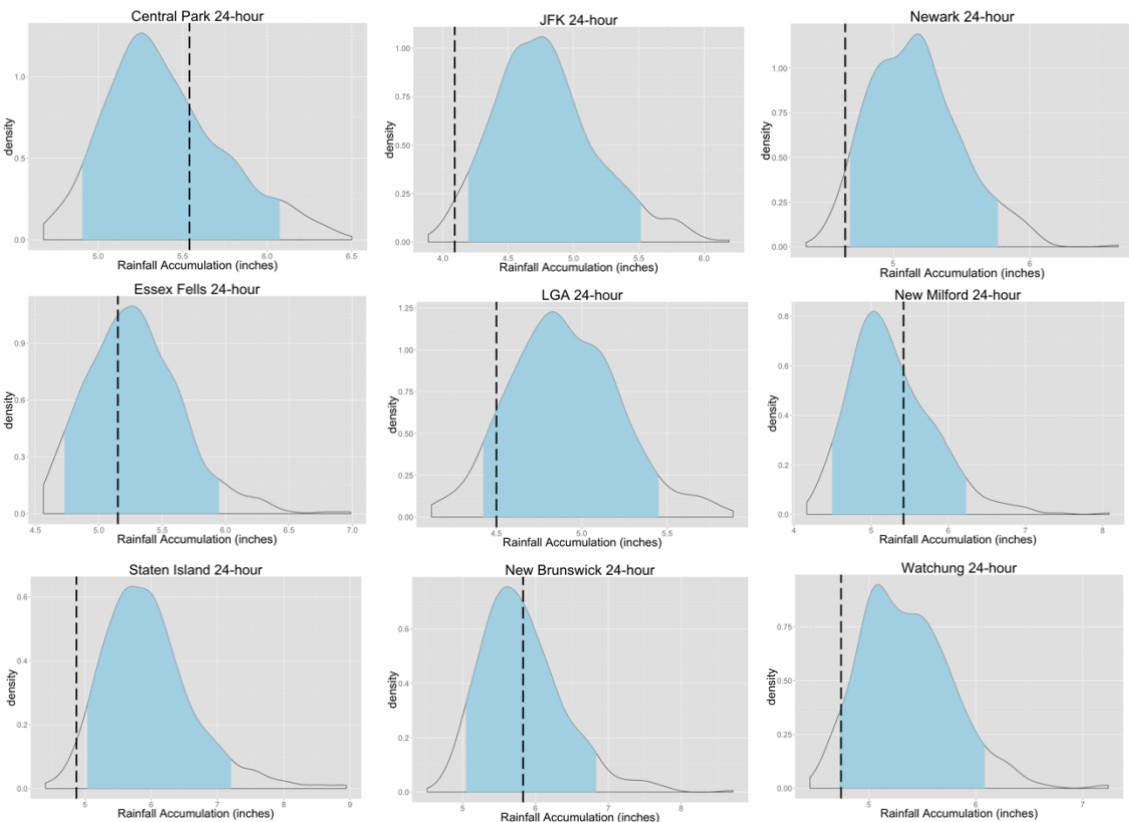

**Figure A3. Density plots of the 24-hr ten-year return period event from SF models for the nine stations (Central Park, Essex Fells, Staten Island – top to bottom left panel; JFK, LGA, New Brunswick - top to bottom, middle panel; Newark, New Milford, Watchung - top to bottom, right panel). The empirical 10-year event is the dotted line, and the 5% - 95% range of simulations is shaded in blue.**