# Peer review of "Technical Note: Modeling Spatial Fields of Extreme Precipitation – A Hierarchical Bayesian Approach"

_Hydrology and Earth System Sciences, 2022_

## Author Comment (AC1)

**Response to Referees**

We thank you for your time and valuable feedback. Please see below, our responses (in blue color text) to the comments.

**Referee 1**

The manuscript "Modeling Spatial Fields of Extreme Precipitation – A Hierarchical Bayesian Approach" by Rahill-Marier et al. introduces a hierarchical Bayesian model for modeling spatial rainfall for extreme events of a specified duration which can be considered in regional hydrologic models to perform a regional hydrologic risk analysis. The spatiotemporal dependence is modeled through multivariate normal with partial pooling for the marginal parameter (mu). The proposed model is used to model the spatial field of rainfall at all 9 stations in New York City. The proposed framework and its application to New York City are interesting and well presented. I have some (minor) comments and technical corrections, especially concerning the ability of the model to capture the spatial dependence structure.

1. I would like to see further analysis about why the empirical quantiles for Staten Island (SI) are consistently underestimated. For example, authors could analyze whether the correlation between SI and other stations is overestimated or not. That could explain the overestimation of the magnitude. In Figure 1, SI shows lighter colors pattern for different durations, which is the opposite of the Central Park pattern.

   Thank you for bringing this up. Upon re-assessment, we think that the comparison we presented here is not one-to-one. Ideally, one has to develop the spatial fields corresponding to specific return period events from the observations also to compare them with the simulations. This, then, will be comparisons of observed extreme spatial fields pooled for specific return periods with the ones from simulated fields from the model and the comparison will be on the probability distribution (or the cumulative distribution) of the observed extreme fields vs. simulated extreme fields.

   We plan to develop and present these comparisons for the revised manuscript.

2. I suspect that it is the first one but would be good if the authors mention which model (partial pooling or no-pooling model). was used to generate figures 3 and 4

   Figures 3, 4 and the A1-A3 (the figures in the appendix) are based on the SF hierarchical (partial pooling) model. We will make this clear in the manuscript and in the figure captions.

3. L129: Event?

   Yes. Thanks for pointing out. It will be corrected in the revised manuscript.

4. L155: The authors say, "Fifty-four models (one for each duration and each site)", but they did not mention which specific duration we consider. Please, specify this.

   We actually run 216 models (9 stations and 24 durations each). We will correct this in the revised manuscript.

5. L158-159: mu and Sigma are leftover.

   We will add the prior assumptions for $\omega_k$ and $\sigma_k^2$ immediately after this sentence.

35  6. L165-166: Authors already mentioned it in L 160-161

    We will delete the redundant sentence in line 165-166 regarding number of iterations per chain.

    7. L193: overestimated?

    You are correct. We will change this in the revised manuscript.

40

---

## Author Comment (AC2)

**Response to Referees**

We thank you for your time and valuable feedback. Please see below, our responses (in blue color text) to the comments.

5 **Referee 2**

Comments on: Technical Note: Modeling Spatial Fields of Extreme Precipitation – A Hierarchical Bayesian Approach by Rahill-Marier et al.

The technical note presented a Hierarchical Bayesian Approach for modelling the spatial fields of extreme precipitation, aiming at simulate rainfall field when extreme rainfall occurring in one or more stations. Hierarchical Bayesian framework is

10 applied for shrinking the rainfall in the space. In general, the technical note is well-written and mathematically rigorous. I am supportive of the publication of the manuscript if the following comments can be addressed.

1. The contribution needs to be highlighted in the introduction. To my understanding, the key difference between a tradition regional rainfall frequency analysis and this spatial fields analysis is the input data, while the Hierarchical Bayesian model is not very different. Thus, it is important to clarify the contribution of this work.

15   We will add the following paragraph towards the end of the introduction.

   "Hierarchical Bayesian models have been applied in the past to spatial fields of annual maximum rainfall. The purpose of these models is to pool information to reduce the uncertainty associated with the return periods of extreme rainfall, and for intensity duration frequency curve analysis. Here, we are interested rather in the spatial distribution of storm rainfall associated with an extreme rainfall event, that is defined such that any location in the region experiences an

20   extreme rainfall event. This is a different goal and informs the rainfall loading a spatially extended drainage network may experience during such an event."

2. For a T-year return period, the return level at a gauge is usually calculated based on the quantile of the annual maxima, which should be independent of the model. In another word, even if calculated with different models or different subsets of data, the return level should be somehow consistent. On line 99, the logged-mean is pooled in the space,

25   where annual maximum (from the target gauge) and non-annual maximum (from other gauges) are pooled. Since the non-annual maximum will usually be smaller than the annual maximum, will the T-year event be systematically underestimated with the approach developed in this technical note? If so, there is a way of overcoming this shortage?

   The traditional approach implicitly considers a T-year event at each site occurring independently. At the event scale, it is rather unlikely that the T-year event estimated using annual maximum data would occur simultaneously at all

30   locations. Thus, from a systems operation perspective, one needs to estimate a model that allows realistic rainfall fields corresponding to extremes to be estimated. To our knowledge, ours is the first approach to consider that. Indeed, it is very likely that for a given event simulated from our model the rainfall at most locations will be smaller than the rainfall experienced at the location with the highest rate. The application of our model would be to draw simulations from the fitted model, such that one or more sites experience a T-year extreme rainfall event, while the other sites

35   experience a rainfall field that is consistent with the pattern to be expected for such an extreme event. The site that gets the extreme event would be simulated randomly in each case for each draw from the model.

3. Line 30, the definition of $R\_djki$ is hard to understand.

40

We use A for annual maximum event for each site and duration and year. Then, based on these events, for each anchor station, we also capture all the rainfall in other stations along with this event. This we denote using R. We will clarify this using a simple example based on a sample year data.

4.   Line 80. The simultaneous fraction analysis showed how many events are concurrent, while it does not provide any information on the spatial dependence of the intensity of extreme rainfall. Although considering spatial structure is important anyway, authors need to make a better justification for this sentence.

45   The counts analysis was developed to summarize the overall fields across all stations. The dependence here is mainly to show if extremes happen simultaneously or not, thereby motivating the need for modelling specific fields for various durations. Once we motivate this, the models have the spatial dependence of the fields constructed through the multi-variate normal distributions on the log-transformed data. We changed the section heading to spatial concordance instead of spatial dependence to clarify.

---

## Author Comment (AC3)

**Response to Referees**

We thank you for your time and valuable feedback. Please see below, our responses (in blue color text) to the comments.

5 **Referee 3**

The present paper explains the break in Clausius-Clapeyron scaling rate in India through use of observation and a surface energy balance approach balance by thermodynamic. The reasoning and the basis of the work is fine, there are few comments I'd like the authors to address before publication (minor revision), as follows:

1. I 'd like to know if the authors have made cross-validation when implementing the method.

10   Since our model is mainly focused on inferring the parameters of the joint distributions and developing return period event, we wanted to use all data for better estimation of the parameters. It was not intended for predictions as there are no time-varying covariates that we are using for estimation. We do agree that if one were to use time-varying covariates, it would be ideal to verify the model in a cross-validated mode.

2. Other than the present assumed distribution of Log-normal, are the GEV or GPD method also tested for comparison?

15   Since not all locations in the spatial fields will have extreme events, we think it is sensible to use the log-normal distribution for estimation of the parameters for the joint distribution of the spatial fields. GEV or GPD are preferred distributions when all the data are block maxima or threshold over values. In our study, it is not that case. Hence, we preferred not set up the models using all extreme value distributions.

3. Line 10 "gaging site" -> "gauging site"

20   4. Line 27 "gage"-> is it gauge?

   NOAA's NCEI (https://www.ncdc.noaa.gov/IPS/hpd/hpd.html) specified these precipitation measuring stations as rain gages. Hence, we used this term. For example:

   *"Description*

25   *This publication contains hourly precipitation amounts obtained from recording rain gages located at*

   *National Weather Service, Federal Aviation Administration, and cooperative observer stations."*

   We will ensure that it is consistent throughout the manuscript in the revised version.

5. Line 30 the annotation is quite hard to understand, I have hard time transforming from R to A. I would expect a
30   simpler annotation used than the one here.

   We use A for annual maximum event for each site and duration and year. Then, based on these events, for each anchor station, we also capture all the rainfall in other stations along with this event. This we denote using R. We will clarify this using a simple example based on a sample year data.

6. Line 83, "across the gauge sites was developed"-> "that was developed"

35

We think this sentence is fine. It was meant to say that a hierarchical Bayesian approach was developed.

> *"A hierarchical Bayesian approach that provides the ability to partially pool model parameters across the rain gauge sites was developed."*